# The Role of Probiotics in Modulating Myocardial Infarction and Depression-like Symptoms: A Study on Sex-Specific Responses

**DOI:** 10.3390/biomedicines12112511

**Published:** 2024-11-02

**Authors:** Marc-André Gagné, Geneviève Frégeau, Roger Godbout, Guy Rousseau

**Affiliations:** 1Research Center, CIUSSS du Nord-de-l’Île-de-Montréal, Hôpital du Sacré-Cœur de Montréal, Montréal, QC H4J 1C5, Canada; marc_andre_96@hotmail.com (M.-A.G.); genevieve.fregeau.cnmtl@ssss.gouv.qc.ca (G.F.); roger.godbout@umontreal.ca (R.G.); 2Department of Pharmacology, Université de Montréal, Montréal, QC H3T 1J4, Canada; 3Department of Psychiatry, Université de Montréal, Montréal, QC H3T 1J4, Canada

**Keywords:** myocardial infarction, reperfusion, probiotics, sex

## Abstract

Background/Objectives: This study explores the effects of two probiotics, *Lactobacillus helveticus* R0052 and *Bifidobacterium longum* R0175, on myocardial infarction (MI) and associated depression-like behaviours, with a focus on sex differences. Methods: MI was induced in adult male and female rats by occluding the left anterior coronary artery for 30 min, followed by 24 h of reperfusion. Probiotics were administered via drinking water for at least two weeks before ischemia. Infarct size, plasma C-reactive protein (CRP), estradiol levels, and intestinal permeability were then measured. Two weeks after, MI subgroups of rats were tested for depression-like behaviours. Results: We found a significant interaction between sex and probiotics in relation to infarct size. Probiotics significantly reduced the infarct size compared to the vehicle group in female rats but not in males. Probiotics increased the plasma estradiol levels and reduced the CRP concentrations in females, suggesting anti-inflammatory and cardioprotective properties. Probiotics significantly increased intestinal resistance following MI in males only, suggesting sex-specific physiological responses to treatment. Probiotics enhanced social interaction in males with MI but not in females. Similarly, in the forced swim test, probiotics reduced immobility in males with MI but increased it in females, further underscoring the sex-dependent effects of probiotics. Conclusions: This study reports cardioprotective effects of probiotics upon MI in female rats, while benefits in male rats were rather at the behavioural level. These results highlight distinct physiological and behavioural responses between sexes, emphasizing the need to account for sex differences in future tests of probiotics as a prophylactic treatment for MI.

## 1. Introduction

A growing body of evidence suggests that probiotics may have therapeutic value across a wide range of medical conditions [1,2]. For instance, in situations where the gastrointestinal system is disrupted—such as in irritable bowel syndrome, inflammatory bowel diseases, and diarrhea [3]—the addition of probiotics has demonstrated beneficial effects. Beyond these gastrointestinal merits, probiotics have also shown the ability to act via the gut–brain axis in experimental settings [4,5,6,7].

Illustratively, our research has demonstrated that administering probiotics to male rats subjected to myocardial infarction resulted in a reduction in depression-related behaviours, a phenomenon frequently observed following a myocardial infarction [8,9]. Indeed, the male rats treated with probiotics showed superior behavioural performance in tests compared to those receiving only a vehicle. Several mechanisms have been proposed to explain these results, including vagus nerve stimulation [10] and reduced inflammation [8]. These observations suggest that the beneficial effects of probiotics can extend beyond the gastrointestinal tract, offering promising prospects for their therapeutic use in broader contexts related to mental health and well-being.

In the realm of cardiovascular diseases, the addition of probiotics can be beneficial under specific conditions. In a myocardial infarction model, it has been shown that probiotics could reduce the size of the myocardial infarction in male rats when subjected to a dysbiotic diet [11]. No protective effect was observed when the animals were on a standard chow diet, suggesting that the beneficial effects of probiotics are more pronounced when the gut microbiota is disrupted.

Despite all these findings, we still do not know if females would benefit from probiotics in a similar manner, especially since the composition of the microbiota varies between sexes [12]. Some studies indicate that the bacteria-to-human cell ratio is higher in women than in men [13] and that women also have greater microbiota diversity. Additionally, studies have shown that ovariectomy can lead to dysbiosis in mice [14]. It is also worth noting that other studies have found that the gut microbiota regulates estrogen levels by modulating its metabolism through enterohepatic circulation [15]. This is particularly intriguing given that estrogens are recognized for their cardioprotective properties. Furthermore, the addition of probiotics has demonstrated that the immune response can be modulated differently according to sex [16], suggesting a potentially different effect on the size of myocardial infarction.

In light of these considerations, this study aims to investigate the effects of two probiotic strains, *Lactobacillus helveticus* Rosell-52 and *Bifidobacterium longum* Rosell-175, on myocardial infarction size and post-infarction depression-related behaviours in female rats. *L. helveticus* R0052 and *B. longum* R0175 have been extensively studied for their impact on the gut–brain axis. These strains have been shown to exert anxiolytic and antidepressant-like effects in animal models and humans by influencing the communication between the gut microbiota and the central nervous system (CNS) [17,18]. Studies suggest that they can modulate the production of neurotransmitters such as serotonin and gamma-aminobutyric acid (GABA), which are crucial for mood regulation [19]. Both strains have been widely studied for their safety and efficacy in human and animal models, making them ideal candidates for this study.

Given the high prevalence of depression-like behaviours following myocardial infarction (MI), the inclusion of these strains is particularly relevant for examining their potential to alleviate psychological stress and depressive symptoms in post-MI conditions. By comparing these effects between sexes, this study seeks to provide insights into the potential sex-specific therapeutic value of probiotics in cardiovascular health and mental well-being.

## 2. Materials and Methods

### 2.1. Ethics Statement

The experiments complied with the animal care guidelines published by the Canadian Council on Animal Care, and the procedures performed (GROU50 and GROU61) were approved by the local Animal Care Committee of the Hôpital du Sacré-Coeur de Montréal (Montréal, QC, Canada). Our study experiments were performed in accordance with the ARRIVE guidelines (Animal Research: Reporting of In Vivo Experiments).

### 2.2. Experimental Design 

A total of 141 adult Sprague Dawley albino rats (approximately 300 g, 72 males and 69 females) (Charles River, Saint-Constant, QC, Canada) were housed in individual cages at 22 °C with a relative humidity of 40–50% and subjected to a 12 h light/dark cycle, starting at 8:00 a.m. The rats had three days of acclimatization to their environment without a specific diet. The animals were randomly assigned to a group using Excel 2019’s random function.

### 2.3. Probiotics Regimen

A blend of freeze-dried cultures of *Lactobacillus helveticus* Rosell-52 and *Bifidobacterium longum* Rosell-175, or the vehicle (maltodextrin), was administered via drinking water. The suspension was freshly prepared daily in 200 mL of tap water. Water intake was monitored to ensure a daily intake of 10⁹ colony-forming units (CFUs) for the probiotics group.

### 2.4. Surgical Procedures

Anesthesia was induced using ketamine (70 mg/kg) and xylazine (10 mg/kg) administered intraperitoneally. The animals were intubated, placed on a respirator, and maintained under anesthesia with 1.5% isoflurane. Throughout the surgery, EKG signals, body temperature, heart rate, and blood oxygen saturation were continuously monitored. The rats were positioned on their right side, and an incision was made in the second intercostal space. The left anterior descending coronary artery was occluded with a 4.0 silk suture and a plastic snare for 30 min. Ischemia was confirmed by ST-segment changes and subepicardial cyanosis. After 30 min, the occlusion was released to allow for reperfusion, and the thoracic cavity was closed. The rats received butorphanol (2 mg/kg) postoperatively and were returned to their cages. After 24 h of reperfusion, the animals were sacrificed by decapitation to measure infarct size, or after 3 weeks for behavioural assessments and scar tissue evaluation. Decapitation was chosen to avoid altering the biochemical pathways.

### 2.5. Infarct Size Measurement

The hearts were harvested immediately after euthanasia, placed on ice, and perfused with saline via the aorta to remove blood. The left anterior descending coronary artery was re-occluded at the same site as during the initial surgery, and the hearts were infused with 0.5% Evans blue solution was used to delineate the area at risk (AR). The hearts were then briefly frozen at −80 °C for 5 min, sliced into four 2 mm transverse sections, and incubated in 1% 2,3,5-triphenyltetrazolium chloride (TTC) solution (pH 7.4) at 37 °C for 10 min to differentiate necrotic tissue (I) from the AR. The left ventricle (LV) areas were traced onto a glass plate, photocopied, cut out, and weighed. Necrosis (I) is expressed as a percentage of the AR (I/AR × 100) and the AR as a percentage of the LV (AR/LV × 100).

### 2.6. Ex Vivo Intestinal Resistance

Small intestine resistance was measured with a Ussing Chamber (Physiologic Instruments, Reno, NV, USA) [11] after 3 weeks of reperfusion. Using a 5 cm segment of jejunum, the mucosa was bluntly stripped from the seromuscular layer and a 1 cm^2^ of mucosa was placed in the cassette. Intestinal transepithelial resistance was measured continuously during the 90 min. The chambers were continuously oxygenated to prevent cell death. The results at 90 min were reported for comparisons.

### 2.7. Plasma Levels of Estradiol and CRP 

Blood was collected in an EDTA-coated tube and centrifuged at 4 °C 1000× *g* for 15 min and the plasma supernatant was kept at −80 °C. An ELISA assay of estradiol (MBS263466, MyBioSource, San Diego, CA, USA) and CRP (DY1744, R&D Systems, Minneapolis, MN, USA) were performed as per the manufacturer’s instruction. The dilution ratio of 1:12,000 was used for the CRP measurement. 

### 2.8. Behavioural Measures

The rats were submitted to 2 behavioural tests 14 days after MI. One test was conducted per day in the morning, individually. The social interaction test and the forced swim test were selected based on their validity regarding depression-like behaviour.

### 2.9. Social Interaction Test

Between 9:00 a.m. and 11 a.m., pairs of rats were placed in a clean shoebox for 10 min. The duration and number of interactions between them were measured for both individuals.

### 2.10. Forced Swim Test

Between 9:00 a.m. and 11 a.m., the rats were placed individually in a transparent 25 cm diameter pool filled with 30 cm of 22–25 °C water, with no possible escape. On the first day, the rats were habituated to the pool for 15 min. The following day, the actual 5 min test was conducted. Observers used identical stopwatches to record the time spent in the 3 different states: immobilized, swimming, and attempting to escape.

### 2.11. Statistical Analysis

The results are expressed as mean ± SEM. The number of animals was calculated based on a range of effect size of 30% from previous studies, alpha errors of 0.05, and power of 0.8. A statistical analysis was performed using a 2-way or 3-way ANOVA with MI, probiotics, and sex as the independent variables. When the interaction was significant, decomposition was performed to detect differences in the simple effect. For the estradiol measurements, the Student *t* test was used. An analysis was performed using SPSS v. 28, and *p* < 0.05 values were considered to be significant. 

## 3. Results

One hundred and forty-one animals were used for this study. Eight died during the myocardial infarction experiment (four females and four males).

### 3.1. Myocardial Infarct Size

The two-way ANOVA analysis reveals a significant interaction between the main factors, sex, and probiotics for the infarct size (F_(1,32)_ = 6.18; *p* < 0.05). The analysis of simple effects indicates that for males, no difference is observed regardless of the presence of probiotics, whereas for females, the difference is significant between females taking probiotics and those receiving the vehicle (Figure 1A; *p* < 0.05). In the presence of probiotics, the infarct size in females is significantly smaller than in the vehicle group. No difference is found in the size of the area at risk between groups.

### 3.2. Estradiol

At 24 h of reperfusion, plasma estradiol concentrations differ in females depending on the presence or absence of probiotics. In the presence of probiotics, estradiol concentrations are higher than those found in females who did not receive probiotics (*p* < 0.05; Figure 2A). No differences are observed between males.

### 3.3. CRP

The measurements of plasma CRP after 24 h of reperfusion indicate a significant interaction between sex and probiotics (F_(1,31)_ = 4.50; *p* < 0.05). Further analysis of this interaction reveals that there is a difference between males and females without probiotics; females have higher levels. The presence of probiotics in females significantly decreases the CRP concentrations after 24 h of reperfusion. No significant differences are observed for the main factors (Figure 2B).

### 3.4. Intestinal Resistance

A three-way ANOVA indicates that the interaction of the three main factors (MI, probiotics, and sex) is not significant, but the interaction between the probiotic and sex (F(_1,77_) = 8.85; *p* < 0.05) and the probiotic and MI (F(_1,77_) = 4.85; *p* < 0.05) are significant. In the first case, without probiotics, there is no difference between sex, but in the presence of probiotics, resistance in males is higher. For the interaction between MI and probiotics, a simple effect analysis indicated that in the absence of MI, the resistance is higher with the probiotics. The main individual factors indicate that females have lower resistance compared to males (F(_1,77_) = 9.15; *p* < 0.05). It also reveals that in the presence of MI, resistance decreases (F(_1,77_) = 5.22; *p* < 0.05) and that probiotics increase intestinal resistance (F(_1,77_) = 5.77; *p* < 0.05; Figure 3).

### 3.5. Scar Extent

A myocardial infarction scar resulting from LAD occlusion manifests a consistent extent in males, regardless of probiotic intake. However, among females, the presence of probiotics significantly reduces the extent of the scar (expressed as % of area at risk) compared to the vehicle group (Figure 1B).

### 3.6. Social Interactions

In males, the passive avoidance test yields noteworthy results, revealing a significant interaction between MI and probiotics (F(_1,71_) = 4.47; *p* < 0.05). Further analysis shows that, within the MI groups, probiotics significantly enhanced social interaction compared to the vehicle group (*p* < 0.05). Additionally, the probiotics and sex interaction is also significant (F(_1,71_) = 8.67; *p* < 0.05). In fact, the presence of probiotics increases the interaction in males (*p* < 0.05), whereas no differences are observed in females. No other significant results are observed (Figure 4).

### 3.7. Forced Swim

A significant interaction emerged between the main factors MI, probiotics, and sex (F(_1,67_) = 5.14; *p* < 0.05). The simple effects of these significant interactions reveal that immobility is superior in the MI/female/probiotics group as compared to the MI/female/vehicle group. In contrast, immobility is reduced in the MI/male/probiotics group as compared to the MI/male/vehicle group. Finally, we also observe a significant effect between the MI/female/probiotics and MI/male/probiotics groups, where the immobility is lower in the male vs. female groups, indicating a sex-dependent effect. The MI main factor indicates a significant difference as immobility is higher in the MI compared to the sham group. No other significant differences are observed (Figure 5).

## 4. Discussion

The results of this study reveal that the administration of probiotics (*L. helveticus* Rosell-52 and *B. longum* Rosell-175) has different effects on the size of the myocardial infarct depending on the sex of the animal. Here, no effects of these probiotics are observed in males, corroborating our previous results [20]. We observed, for the first time, a decrease in the size of the infarct in females pretreated with these probiotics. This finding is supported by the scar size in the females from group 2, which is also smaller in the probiotics-treated animals, indicating a difference in ischemia/reperfusion-induced myocardial damage.

At first glance, this sex-dependent cardioprotective effect could be linked to the presence of female hormones that have demonstrated cardioprotective effects. Indeed, several studies in animals have shown that 17ß-estradiol possesses cardioprotective properties. For instance, an infusion of 17ß-estradiol into isolated heart preparations [21] significantly decreases the extent of damage in both male and female isolated hearts, potentially through a mechanism involving mitochondrial protection. Other studies also reported a reduction in infarct size with estradiol administration [22,23], although these conclusions are challenged in the presence of a permanent occlusion [24]. Activation of the G protein-coupled receptor 30 that binds to estrogen offers protection by preventing the opening of the mitochondrial permeability transition pore (mPTP). During ischemia and especially during reperfusion, there is a large increase in reactive oxygen species (ROS) production and calcium influx into cells. These factors can cause the mPTP to open, which leads to a loss of mitochondrial membrane potential, the release of pro-apoptotic factors like cytochrome c, and subsequent cell death [25].

In our study, we observe that the presence of probiotics increases plasma estradiol concentration in females. It has long been recognized that the intestinal microbiota can influence estrogen levels, whether after antibiotic treatment [26] or in germ-free mice [27]. Although the mechanism remains unknown, some have suggested that the microbiome may produce ß-glucuronidases, which can block estrogen binding to glucuronic acid, thereby reducing estrogen inactivation [28]. Another hypothesis suggests the involvement of the gut–brain axis [29], although the mechanism remains highly speculative at this time [29].

Another difference observed in response to probiotics is in the plasma concentration of C-reactive protein (CRP), which shows significant variations based on sex. In the present study, no changes in CRP levels are observed in males in the presence of probiotics, but females demonstrate lower CRP levels at 24 h of reperfusion. These findings align with a meta-analysis indicating that probiotics can reduce serum CRP concentrations [30], a marker of elevated systemic inflammation commonly associated with various pathological conditions, such as cardiovascular diseases. Researchers have hypothesized that the anti-inflammatory effects of certain probiotic strains, including those used in our study [31], may contribute to the reduction in CRP concentration [32,33]. Furthermore, accumulating evidence suggests sex-specific effects of probiotics [34,35], implying that, in our model, this particular combination of probiotics appears to be cardioprotective for females but not for males. Numerous studies demonstrate an inverse relationship between estradiol levels and CRP concentrations. Higher levels of estradiol, such as those we observe in the probiotic group, are generally associated with lower CRP levels, indicating a reduced inflammatory state [36]. Estradiol reduces inflammation by down-regulating the production of pro-inflammatory cytokines (such as TNF-α and IL-6), which are known to stimulate CRP production in the liver. By inhibiting these cytokines, estradiol can lower CRP levels, thereby reducing the overall inflammatory burden in the body. 

It is interesting to note that intestinal barrier resistance is significantly higher in males than in females in the presence of probiotics and that myocardial infarction tends to reduce it (Figure 3). A reduction in this resistance can promote the translocation of bacterial products, such as lipopolysaccharides, into the bloodstream, contributing to an increase in systemic inflammation [37,38]. This dynamic could also explain why post-infarction inflammation is more pronounced in females. However, this difference in resistance does not seem to directly explain the effect on infarct size but may play a more important role in the mechanisms involved in depression-like behaviour [39,40]. Although the tested probiotics appear to have a more pronounced effect in males, the current experimental design does not allow us to fully understand the underlying reasons for this sex difference.

In previous studies, we observed that the induction of myocardial ischemia followed by reperfusion in male rats leads to symptoms similar to those found in depression [41]. Indeed, myocardial infarction is associated with an increased incidence of developing depression in both humans and rats [41,42]. Thus, by subjecting animals to behavioural tests to assess their depressive state, we observed that in the socialization test, male animals under probiotics interacted more with their counterparts than male animals under the vehicle [9].

However, this result was not replicated in females, as the socialization time remains similar with or without probiotics. This finding suggests that myocardial infarction and our probiotic mixture do not appear to have an influence on socialization in females. In their research, Carrier and Kabbaj [43] observe that female rats exhibit a comparatively diminished sensitivity in responding to the social interaction test when compared to males. This discrepancy implies a heightened level of anxiety among females, potentially accounting for the observed sex differences in test outcomes.

Similarly, both male and female subjects display increased immobility in the forced swim test when confronted with infarction. While we observed that males with myocardial infarction (MI) are less immobile in the forced swim test when exposed to probiotics compared to the vehicle, suggesting a positive effect on the depressive state induced by the MI, the opposite is observed in females. These results further demonstrate that the probiotic mixture we used induces sex-dependent effects. 

In summary, our research indicates that the effects of probiotic consumption differ between males and females. The administration of these probiotics to females results in a reduction in infarct size, highlighting a distinct impact on females compared to males. However, probiotics have positive effects in reducing depression-like symptoms in males after a myocardial infarction but not in females.

The sex-specific effects of probiotics observed in this study have important implications for future research and clinical practices. Understanding the mechanisms by which probiotics influence cardiovascular outcomes and mental health differently in males and females can guide the development of more personalized probiotic therapies. For instance, probiotics could be explored as a potential cardioprotective treatment specifically for women, while their benefits in mental health might be further developed for men post-MI. Our findings pave the way for future studies aimed at elucidating the underlying mechanisms behind these sex-dependent effects, ultimately contributing to more effective and targeted therapeutic strategies in both cardiovascular and mental health fields.

## Figures and Tables

**Figure 1 biomedicines-12-02511-f001:**
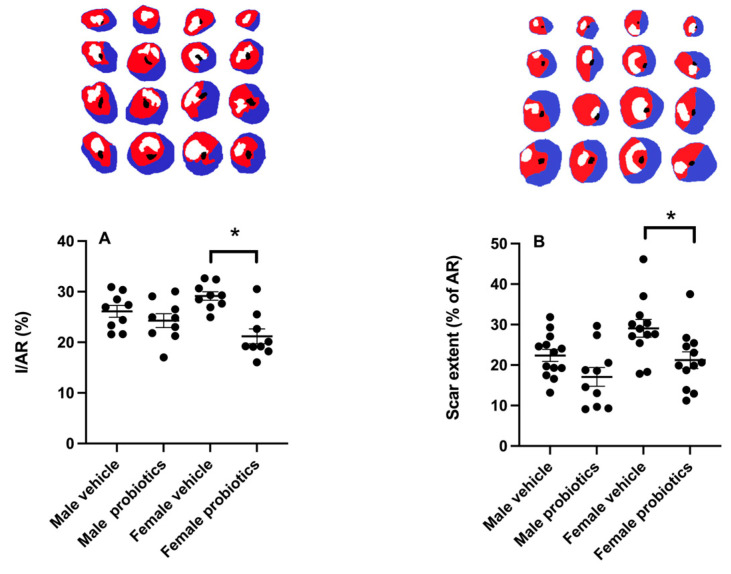
(**A**) Infarct size (I) expressed as percent of the area at risk (AR) after 24 h reperfusion. (**B**) Scar extent expressed as percent of the AR after >3 weeks post-ischemia. Two-way ANOVA was used to determine the difference between groups. * *p* < 0.05 between groups.

**Figure 2 biomedicines-12-02511-f002:**
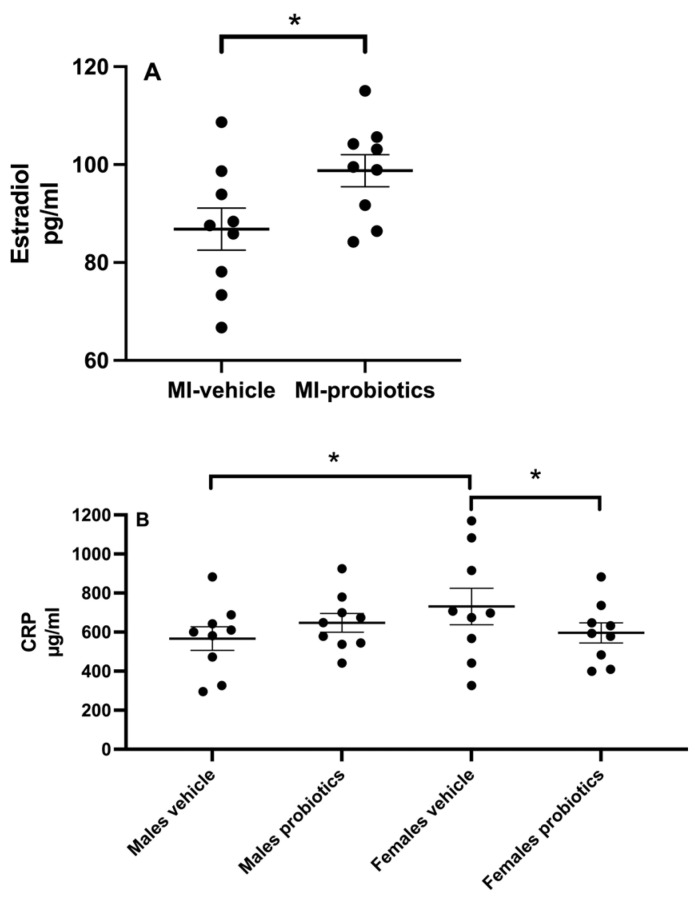
(**A**)—Plasmatic concentrations of estradiol observed in females 24 h after the induction of ischemia. Statistical analysis was performed using Student *t* test.* *p* < 0.05 between groups. (**B**)—CRP plasmatic concentrations of CRP observed in MI animals, 24 h after the induction of myocardial ischemia. Two-way ANOVA was used to determine the difference between groups. * *p* < 0.05 between groups.

**Figure 3 biomedicines-12-02511-f003:**
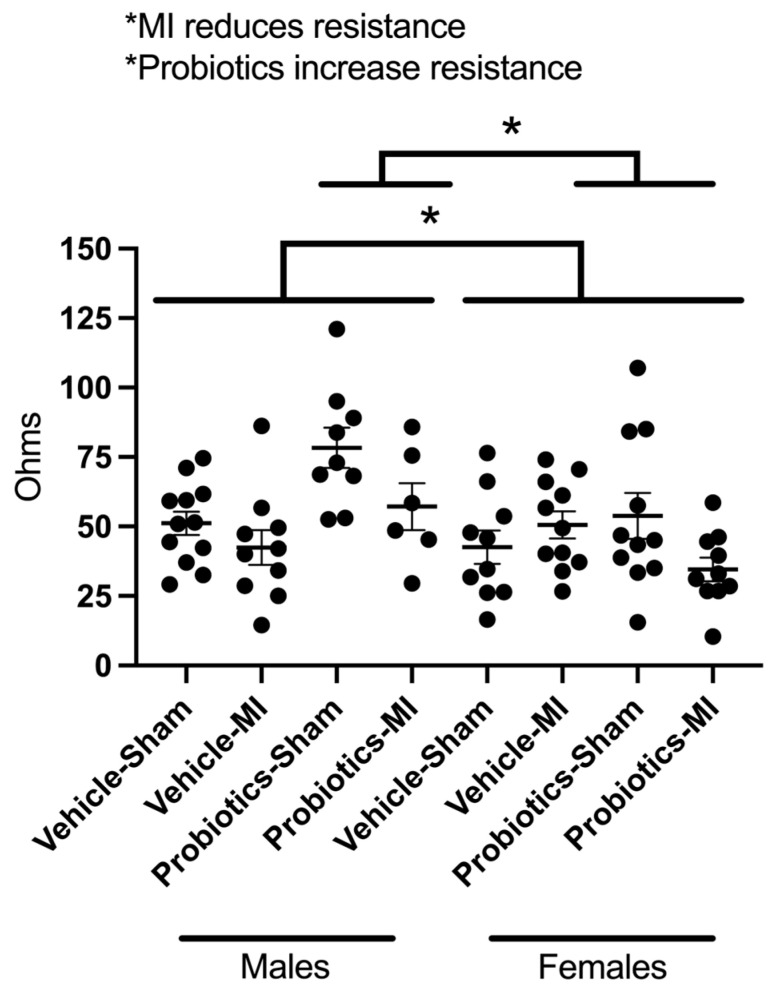
Intestinal resistance gut barrier resistance (ohms) measured at 3 weeks post-MI in the different groups with the Ussing Chambers. Three-way ANOVA was used to determine the difference between groups. * *p* < 0.05 for the MI, probiotics, and sex main factors. The interaction between MI and probiotics and probiotics and sex are significant.

**Figure 4 biomedicines-12-02511-f004:**
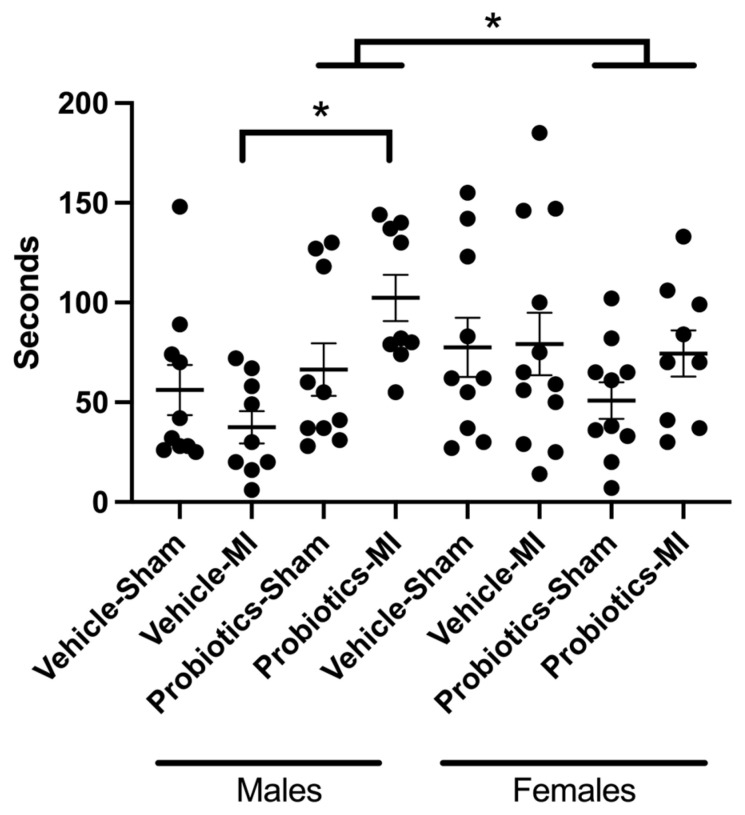
Social interaction between animals. Three-way ANOVA was used to determine the difference between groups. Interactions between animals measured in seconds in the different experimental groups. * *p* < 0.05 for the interactions between probiotics and sex and MI and probiotics.

**Figure 5 biomedicines-12-02511-f005:**
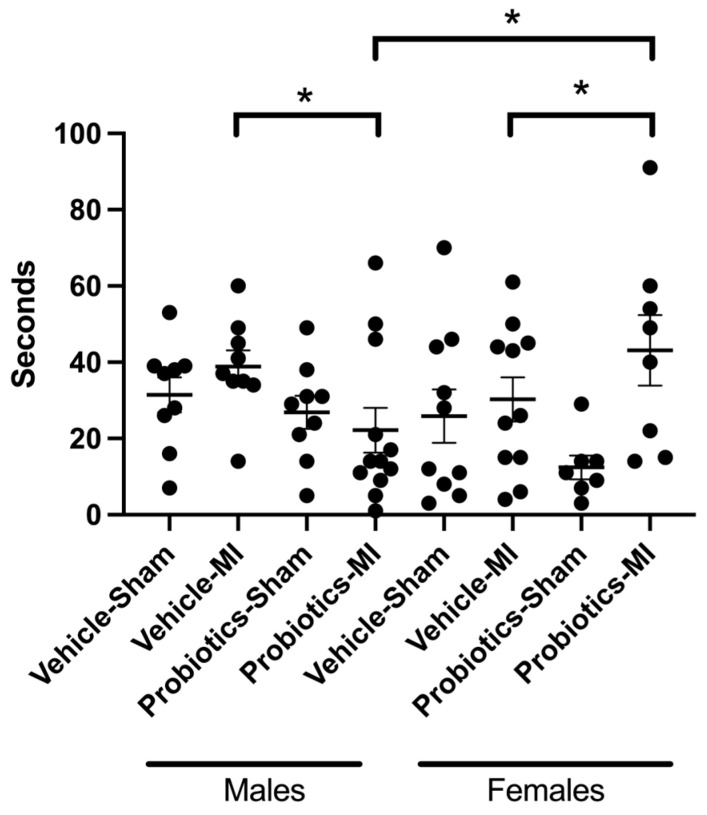
Immobility in the forced swim test. Immobility measured in seconds of rats from the different groups in the forced swim test. Three-way ANOVA was used to determine the difference between groups. * *p* < 0.05 for the interaction between the three main factors (MI, probiotics, and sex).

## Data Availability

The raw data supporting the conclusions of this article will be made available by the authors on request.

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
