# Peer review of "The Role of Probiotics in Modulating Myocardial Infarction and Depression-like Symptoms: A Study on Sex-Specific Responses"

_biomedicines, 2024, doi:10.3390/biomedicines12112511_

Round 1

Reviewer 1 Report

Comments and Suggestions for Authors

Comments to the Author

The manuscript presents an interesting and relevant investigation into the cardioprotective effects of probiotics, specifically Lactobacillus helveticus R0052 and Bifidobacterium longum R0175, on myocardial infarction (MI) and associated depression-like behaviors, with a focus on sex differences. The research addresses an important gap in understanding the role of probiotics in cardiovascular health and the potential sex-specific responses. However, some aspects of the manuscript need improvement for clarity, structure, and methodological transparency.

1.       The abstract of this article includes the phrase: "Abstract: The abstract should be a total of about 250 words and structured to contain the following headings: Background/Objectives, Methods, Results, Conclusions." What does this mean?

2.       The Abstract should be completely rephrased since it is confused and doesn't well represent the paper content.

3.       Please clearly articulate the contribution of your paper in the Introduction section. Highlight the key findings and explain how they advance current knowledge in the field. Emphasize the unique aspects of your research and the specific gaps it addresses. This will help readers quickly understand the importance and relevance of your study.

4.       Please describe the potential implications of your work for the field and how it might advance knowledge or improve clinical practices. This will help reviewers appreciate the direction and significance of your future work.

5.     The paper contents are a little bit confused and not well organized.

6.       The authors rationale for selecting the probiotic strains Lactobacillus helveticus R0052 and Bifidobacterium longum R0175 is insufficient, and the cited references are outdated. It is recommended to consider reviewing the following article:

Yanting Han, 2024.  https://pubs.acs.org/doi/10.1021/acs.biomac.4c00124.

Yin, Z.,2024. DOI:10.1097/nr9.0000000000000055

7.       The article identifies sex-specific differences; however, the discussion lacks an in-depth analysis of the underlying mechanisms. In particular, the relationship between increased estradiol levels and reduced myocardial infarction size has not been sufficiently explored, including the role of estradiol and other potential factors such as inflammation regulation and metabolic differences.

8.       In the results section, the relationship between increased intestinal resistance and myocardial infarction is mentioned, but the biological significance of this finding is not clearly explained. The mechanism behind this result remains unclear, and further clarification is needed to establish the logical connection between increased intestinal resistance, myocardial infarction, and the role of probiotics.

Comments on the Quality of English Language

Make some modifications and refinements to the article to enhance the fluency and accuracy of the language. It is advisable for the authors to conduct a thorough language check before final submission to ensure there are no grammatical or spelling errors.

Author Response

Reviewer #1

  1. The abstract of this article includes the phrase: "Abstract: The abstract should be a total of about 250 words and structured to contain the following headings: Background/Objectives, Methods, Results, Conclusions." What does this mean?

Answer: We apologize for this mistake. The phrase was included in error and has been removed in the revised version. The abstract is now correctly formatted according to the required structure.

  1. The abstract should be completely rephrased since it is confused and doesn't well represent the paper content.

Answer: The abstract has been completely rephrased.

  1. Please clearly articulate the contribution of your paper in the Introduction section. Highlight the key findings and explain how they advance current knowledge in the field. Emphasize the unique aspects of your research and the specific gaps it addresses. This will help readers quickly understand the importance and relevance of your study.

Answer: Thank you for this valuable suggestion. In the revised version, we have enhanced the Introduction section by clearly articulating the key contributions of our paper. We have highlighted the main findings of our research and explained how they advance current knowledge in the field. We have also emphasized the unique aspects of our studies and specified the research gaps it addresses, ensuring that readers can easily grasp the importance and relevance of our work.

  1. Please describe the potential implications of your work for the field and how it might advance knowledge or improve clinical practices. This will help reviewers appreciate the direction and significance of your future work.

Answer: Thank you for your valuable suggestion. In response, we have expanded the discussion to outline the potential implications of our work. Our findings highlight sex-specific effects of probiotics on both cardiovascular and behavioural outcomes, which could pave the way for more personalized probiotic therapies in clinical settings. This research advances knowledge by demonstrating that probiotics may offer targeted benefits based on sex, potentially improving treatment strategies for conditions like myocardial infarction and related mood disorders. These insights could guide future clinical practices by encouraging a more individualized approach to probiotic use, considering patient sex as a critical factor in treatment efficacy. We plan to explore these aspects further in subsequent studies to strengthen the evidence for tailored probiotic interventions.

  1. The paper contents are a little bit confused and not well organized.

Answer: We are sorry to hear this feedback. While we have followed the journal's guidelines, we have made additional efforts in the revised version to improve the clarity and organization of the content for better readability.

  1. The authors’ rationale for selecting the probiotic strains Lactobacillus helveticus R0052 and Bifidobacterium longum R0175 is insufficient, and the cited references are outdated. It is recommended to consider reviewing the following article:

Yanting Han, 2024.  https://pubs.acs.org/doi/10.1021/acs.biomac.4c00124.

Yin, Z.,2024. DOI:10.1097/nr9.0000000000000055

Answer: While we understand your comment, we respectfully disagree. In the case of probiotics, it is well known that the specific species can greatly influence the response. Therefore, it is crucial that the references pertain to the same species in order to make valid comparisons. Additionally, although some of the cited articles may be older, they remain relevant as the probiotics used are the same as those in our current study.

  1. The article identifies sex-specific differences; however, the discussion lacks an in-depth analysis of the underlying mechanisms. In particular, the relationship between increased estradiol levels and reduced myocardial infarction size has not been sufficiently explored, including the role of estradiol and other potential factors such as inflammation regulation and metabolic differences.

Answer: In agreement with the comments from the other reviewers, we have added a section to the discussion that explores the potential link between inflammation, probiotics, and sex-specific differences. We have also expanded on the role of estradiol, particularly its influence on inflammation regulation and metabolic differences, and how these factors may contribute to the observed reduction in myocardial infarction size in females.

  1. In the results section, the relationship between increased intestinal resistance and myocardial infarction is mentioned, but the biological significance of this finding is not clearly explained. The mechanism behind this result remains unclear, and further clarification is needed to establish the logical connection between increased intestinal resistance, myocardial infarction, and the role of probiotics.

Answer: Thank you for pointing this out. We agree that the connection between increased intestinal resistance and myocardial infarction requires further clarification. In the revised version, we have expanded on the biological significance of this finding in the discussion. Specifically, we explore potential mechanisms, including how increased intestinal barrier function could reduce systemic inflammation, which is a known contributor to myocardial infarction. We also delve into the role of probiotics in enhancing intestinal resistance and how this could indirectly protect the heart by mitigating inflammatory pathways. By doing so, we aim to provide a clearer and more logical connection between these factors.

Reviewer 2 Report

Comments and Suggestions for Authors

The study investigated the impact of two probiotic strains, Lactobacillus helveticus R0052 and Bifidobacterium longum R0175, on myocardial infarction and depression-like symptoms in rats, with a specific focus on sex differences. The study demonstrates significant sex-specific effects of probiotics on myocardial infarct size, scar extent, estradiol levels, and inflammatory markers, with marked differences between male and female rats. However there remained some concerning issues.

1. For the infarct size evaluation, the authors should at least use echocardiography to evaluate the LVEF. Meanwhile, the authors should provide Evans blue stained images.

2. The authors should indicate significant difference with *in Figure 3, 4, 5

3. Plasmatic concentrations of CRP are higher in female-vehicle group compared to male-vehicle group. The function of probiotics on female CRP might be resulted to higher CRP level in female-vehicle group. How do you interpret the results.

4. Spelling errors throughout the manuscript should be revised.

Comments on the Quality of English Language

Moderate editing of English language required.

Author Response

Reviewer #2

  1. For the infarct size evaluation, the authors should at least use echocardiography to evaluate the LVEF. Meanwhile, the authors should provide Evans blue stained images.

Answer: For the evaluation of infarct size, we used gross macroscopic measurement, which offers a direct and reliable assessment. While echocardiography is useful for assessing LVEF and overall cardiac function, it does not directly quantify infarct size, which is why LVEF was not included in this study. Additionally, we have added a figure showcasing a representative example of tissue staining for each group to enhance visual clarity.

  1. The authors should indicate significant difference with “*” in Figure 3, 4, 5

Answer: Since a factorial ANOVA was used for the analysis, it can be challenging to clearly differentiate between interactions and main effects using a simple ‘*’ notation. However, we have made our best effort to clarify this by providing detailed statistical results and visual representations that highlight significant interactions and main effects more clearly.

  1. Plasmatic concentrations of CRP are higher in female-vehicle group compared to male-vehicle group. The function of probiotics on female CRP might be resulted in higher CRP level in female-vehicle group. How do you interpret the results.

Answer: The observation of higher plasmatic CRP concentrations in the female-vehicle group compared to the male-vehicle group aligns with existing literature suggesting that females often exhibit higher baseline CRP levels due to hormonal and physiological differences, particularly the influence of estrogen. The effect of probiotics on female CRP levels may be influenced by this baseline elevation, which could explain the differential response between genders. Further investigations into sex-specific responses to probiotics and CRP regulation are warranted to clarify these differences and their underlying mechanisms. We add text in the discussion section.

  1. Spelling errors throughout the manuscript should be revised.

Answer: We apologize for these errors and have made every effort to correct them in this revised version.

Round 2

Reviewer 2 Report

Comments and Suggestions for Authors

I have no more comments

Comments on the Quality of English Language

Minor editing of English language required.

Author Response

Minor editing of language required: The text has been reviewed and edited by two native English speakers to address minor language issues.